# Combined Use of Presepsin and (1,3)-β-D-glucan as Biomarkers for Diagnosing *Candida* Sepsis and Monitoring the Effectiveness of Treatment in Critically Ill Patients

**DOI:** 10.3390/jof8030308

**Published:** 2022-03-17

**Authors:** Radim Dobiáš, Marcela Káňová, Naděžda Petejová, Štefan Kis Pisti, Robert Bocek, Eva Krejčí, Helena Stružková, Michaela Cachová, Hana Tomášková, Petr Hamal, Vladimír Havlíček, Milan Raška

**Affiliations:** 1Department of Bacteriology and Mycology, Public Health Institute in Ostrava, 702 00 Ostrava, Czech Republic; radim.dobias@zuova.cz (R.D.); eva.krejci@zuova.cz (E.K.); michaela.cachova@zuova.cz (M.C.); 2Institute of Laboratory Medicine, Faculty of Medicine, University of Ostrava, 703 00 Ostrava, Czech Republic; helcastruzkova@seznam.cz; 3Department of Anesthesiology and Intensive Care Medicine, University Hospital Ostrava, 708 00 Ostrava, Czech Republic; 4Institute of Physiology and Pathophysiology, Faculty of Medicine, University of Ostrava, 710 00 Ostrava, Czech Republic; 5Department of Intensive Medicine, Emergency Medicine and Forensic Studies, University of Ostrava, 710 00 Ostrava, Czech Republic; 6Department of Internal Medicine, University Hospital Ostrava, 708 00 Ostrava, Czech Republic; petejova@seznam.cz; 7Department of Internal Medicine, University of Ostrava, 703 00 Ostrava, Czech Republic; 8Department of Internal Medicine III—Nephrology, Rheumatology and Endocrinology, Faculty of Medicine and Dentistry, Palacký University Olomouc, 775 15 Olomouc, Czech Republic; 9Department of Anesthesiology and Resuscitation, Ostrava City Hospital, 728 80 Ostrava, Czech Republic; kispisti@seznam.cz; 10Department of Anesthesiology and Resuscitation, Hospital with Polyclinic Havířov, 736 01 Havířov, Czech Republic; robert.bocek@nsphav.cz; 11Center of Health Services, Institute of Public Health in Ostrava, 702 00 Ostrava, Czech Republic; hana.tomaskova@zuova.cz; 12Department of Epidemiology and Public Health, Faculty of Medicine, University of Ostrava, 703 00 Ostrava, Czech Republic; 13Department of Microbiology, Faculty of Medicine and Dentistry, Palacký University Olomouc, 775 15 Olomouc, Czech Republic; petr.hamal@fnol.cz; 14Institute of Microbiology of the Czech Academy of Sciences, 142 20 Prague, Czech Republic; vlhavlic@biomed.cas.cz; 15Department of Analytical Chemistry, Faculty of Science, Palacký University, 771 46 Olomouc, Czech Republic; 16Department of Immunology, Faculty of Medicine and Dentistry, Palacký University Olomouc, 775 15 Olomouc, Czech Republic; milan.raska@upol.cz

**Keywords:** sepsis, *Candida*, bloodstream infections, presepsin, procalcitonin, C-reactive protein, (1,3)-β-D-glucan

## Abstract

New biomarker panel was developed and validated on 165 critically ill adult patients to enable a more accurate invasive candidiasis (IC) diagnosis. Serum levels of the panfungal biomarker (1,3)-β-D-glucan (BDG) and the inflammatory biomarkers C-reactive protein, presepsin (PSEP), and procalcitonin (PCT) were correlated with culture-confirmed candidemia or bacteremia in 58 and 107 patients, respectively. The diagnostic utility was evaluated in sensitivity, specificity, positive predictive value (PPV), and negative predictive value (NPV). BDG was the best marker for IC, achieving 96.6% sensitivity, 97.2% specificity, 94.9% PPV, and 98.1% NPV at a cut-off of 200 pg/mL (*p* ≤ 0.001). PSEP exhibited 100% sensitivity and 100% NPV at a cut-off of 700 pg/mL but had a lower PPV (36.5%) and low specificity (5.6%). Combined use of PSEP and BDG, thus, seems to be the most powerful laboratory approach for diagnosing IC. Furthermore, PSEP was more accurate for 28-day mortality prediction the area under the receiver operating characteristic curve (AUC = 0.74) than PCT (AUC = 0.31; PCT cut-off = 0.5 ng/mL). Finally, serum PSEP levels decreased significantly after only 14 days of echinocandin therapy (*p* = 0.0012). The probability of IC is almost 100% in critically ill adults with serum BDG and PSEP concentrations > 200 pg/mL and >700 pg/mL, respectively, defining a borderline between non-invasive superficial *Candida* colonization and IC.

## 1. Introduction

Critically ill patients are at high risk of hospital-acquired infection, with invasive candidiasis (IC) being among the most common diseases. IC usually occurs after a prolonged ICU stay [1], and in recent decades, infections of non-albicans *Candida* species have become more prevalent than *Candida albicans* infections [2]. Candidemia remains associated with high mortality and treatment costs worldwide [3]; the 28-day ICU mortality is about 44% [4]. Indwelling devices, including central venous catheters (CVCs), are of particular concern as sources of *Candida* infection. Echinocandins are the drugs of choice for treating IC, and antifungal treatment should continue for at least 14 days until blood cultures confirm the resolution of fungemia. Susceptibility testing is crucial for detecting resistance and managing the transition to oral treatment. In persistent candidemia, echocardiography is necessary, and ophthalmoscopy should be considered [5].

The dynamics of the serum concentrations of the panfungal antigen 1,3-β-D-glucan (BDG) enable differentiation between bacterial and fungal infections. However, the clinical significance of elevated BDG concentrations in the early stages of IC in critically ill patients is unclear. There are no widely accepted cut-off BDG concentrations for distinguishing bacteremia from candidiasis [6]. Therefore, the discriminative and predictive value of various serum BDG cut-off concentrations for diagnosing *Candida* spp. colonization, catheter sepsis, invasive candidiasis, and distinguishing deep-seated candidiasis from bacteremia were analyzed.

The potential for diagnosing candidiasis using three early inflammatory biomarkers of C-reactive protein (CRP), procalcitonin (PCT), and presepsin (PSEP) were also evaluated. While the limitations of CRP are well known, PCT is a valuable biomarker in the initial stages of bacterial infections because its serum concentration in Gram-positive and fungal infections is lower than in Gram-negative infections. Accordingly, its serum concentrations are deficient (2–3 ng/L) in the IC [7,8]. PSEP, a soluble CD14 protein, is a highly accurate predictor of sepsis according to the Sepsis-3 criteria [9,10], and it has good diagnostic accuracy for bacteremia and candidemia, including the IC [11]. For this reason, the predictive values of combinations of PSEP, PCT, CRP, and BDG as markers for diagnosing *Candida* sepsis and monitoring the efficacy of echinocandin therapy were analyzed. 

## 2. Materials and Methods

### 2.1. Study Design, Patient Selection, and Outcome

This multicenter retrospective observational cohort study included patients aged > 18 years admitted to ICUs in three centers (University Hospital Ostrava, City Hospital Ostrava, Havířov Hospital) between January 2018 and December 2020. Their enrollment was based on clinical suspicion of sepsis due to severe organ dysfunction, indicated by an increase in the Sequential Organ Failure Assessment Score (SOFA, sepsis-related) of 2 or more, together with a confirmation of IC following the European Organization for Reseasrch and Treatment of Cancer and the Mycoses Study Group Education and Research Consortium (EORTC/MSGERC) consensus definitions [12]. Data recorded for individual patients included: a reason for ICU admission, age, sex, surgical, traumatic, or medical patient type, *Candida* colonization (indicated by one or more *Candida*-positive cultures from non-sterile sites), SOFA score [13,14], Acute Physiology and Chronic Health Evaluation II score [15,16], serum creatinine concentration, and information about the following binary variables: diabetes mellitus status, Central Venous Catheter (CVC) introduced for more than 48 h, renal replacement, and corticosteroid therapy. Neutropenic and transplanted patients were excluded due to the minimal size of the respective cohort. The study was part of a registered clinical trial evaluating biomarkers for early diagnosis of sepsis in patient groups with culture-confirmed candidemia or bacteremia to identify suitable cut-off concentrations for a biomarker panel.

The sera for analysis of the following biomarkers were collected: (a) for BDG within 48 h before the time of the indexed blood culture (IBC), and (b) for PCT, CRP, and PSEP within 24 h before IBC. The time windows for the markers differed because of their blood kinetics [17,18]. Kinetics data for PSEP are summarized in the Appendix A. The time of the index culture corresponds to the first blood culture positivity for *Candida* spp. or the first detection of bacteria by the BacT/ALERT® automated culture system (bioMérieux, France). Episodes of candidemia or bacteremia, where the data (biomarker versus IBC more than 48 hours apart) did not overlap according to the kinetics of biomarkers described above. In the included patients on the echinocandin therapy group, the significance of serum PSEP and BDG concentrations reductions was assessed 3, 14, and 28 days after therapy initiation.

### 2.2. Samples and Laboratory

BDG, PSEP, PCT, CRP, and creatinine levels were measured in accredited laboratories of the Public Health Institute in Ostrava and participating hospitals. BDG measurements were performed using the Fungitell assay (assay range < 0.07–2197 pg/mL, Associates of Cape Cod, Inc., East Falmouth, MA, USA). PSEP (assay range 15–30,000 pg/mL) and PCT (assay range 0.02–75 ng/L) were measured using a quantitative sandwich chemiluminescence immunoassay (Pathfast, Mitsubishi, Japan, and ADVIA Centaur, BRAHMS PCT Assay, Siemens, Munich, Germany). CRP (assay range 1–480 mg/L) was measured using the CRP Latex test (Beckman Coulter, Breya, CA, USA).

*Candida* spp. and bacteria were identified by MALDI-TOF mass spectrometry (Bruker Daltonics, Bremen, Germany).

### 2.3. Statistical Analysis

Descriptive statistics (medians, arithmetic means, standard deviations, and minimum and maximum values) and frequency tables characterize the study groups. Significance was evaluated using the non-parametric two-sample Wilcoxon test, chi-square test, and Fisher’s exact test. Several methods for cut-off value estimates (maximize spline, gam, loess estimates for metrics accuracy and sum of sensitivity and specificity, and Kernel estimate for Youden index) were used. 

Sensitivity, specificity, PPV, NPV, and accuracy were calculated using 95% confidence intervals and (AUC). A significance threshold of *p* < 0.05 was applied in all cases. All statistical analyses were performed using STATA version 13.

## 3. Results

### 3.1. Patient Characteristics and Outcome

A total of 165 patients who tested positive for candidemia (n = 58) or bacteremia (n = 107) were included in the study. PKatients with positive yeast cultures (except for sputum) within seven days before IBC to minimize the effect of *Candida* colonization and patients with the mixed yeast and bacteria blood cultures were excluded. In 17 out of 58 candidemia patients, consecutive measurements of biomarkers were performed on days 3, 14, and 28 after initiating echinocandin therapy (Figure 1). The patients’ baseline clinical characteristics are summarized in Table 1. The presence of a CVC or peripherally inserted central catheter (PICC) was associated with candidemia more often than with bacteremia (100% vs. 69%, *p* < 0.001), and the same was true for administration of antibiotics (83% vs. 42%, *p* < 0.001). Candidemia developed later after hospital admission than bacteremia (23 vs. 13 days, *p* < 0.001) (Table 1).

### 3.2. PSEP Exhibits the Highest Predictive Value for 28-Day Mortality

Although there was no significant difference in mortality between candidemia (33%) and bacteremia (22%) patients (*p* = 0.141) 28 days after initiating therapy, a two-sample Wilcoxon rank-sum test analysis using four biomarkers (CRP, PCT, PSEP, and BDG) in conjunction with the SOFA score (*p* = 0.012) revealed significant differences between the two patient groups (Appendix A). PSEP correlated significantly with PCT, and multivariable logistic regression analysis showed that PSEP (AUC = 0.74) had a significantly higher (*p* = 0.013) predictive value for mortality than PCT (AUC = 0.31) in IC cases (Appendix A). The sensitivity and specificity of PSEP as a predictor of 28-day mortality were 81% and 69%, respectively; the corresponding values for PCT were 68% and 23%, respectively.

### 3.3. Serum CRP, PCT, PSEP, and BDG for Differentiation between Candida and Bacterial Sepsis

CRP, PCT, PSEP, and BDG serum levels were determined in blood culture-positive patients (candidemia n = 58, bacteremia n = 107) (Figure 1). The bacterial species identified in the cultures are listed in the legend of Table 1. The predictive utility and differential diagnosis capacities of these biomarkers are summarized in Table 2. Cut-off concentrations of 5 mg/ml for CRP and 350 pg/mL for PSEP were not useful for discriminating between candidemia and bacterial sepsis because the serum concentrations of these markers exceeded these cut-offs in all patients in both groups. Similarly, no statistically significant discrimination was achieved using PCT with a cut-off of 0.5 ng/L (*p* = 0.406). Conversely, PCT with a 3 ng/L cut-off and CRP with a 130 mg/mL cut-off were moderately discriminative (*p* = 0.012 and *p* < 0.001, respectively), giving positive results for 76% and 68% of candidemia patients, respectively, compared to 56% and 38% of bacteremia patients, respectively. The most discriminative single biomarker was BDG with an 80 pg/mL cut-off: this threshold value was exceeded in 58/58 candidemia patients (100%), but in only 19/107 bacteremia patients (18%; *p* < 0.001). The median serum BDG concentration was significantly higher in candidemia (1029 pg/mL) than in bacteremia (35 pg/mL; *p* < 0.001). When the BDG cut-off was increased to 200 pg/mL (or 220 pg/mL), the discrimination between the candidemia and bacteremia groups increased. Still, the percentage of BDG-positive candidemia patients fell from 100% to 97% (or 95%). Therefore, we tested whether combinations of these serum biomarkers could achieve high discrimination without reducing sensitivity to candidemia.

In summary, the individual biomarkers of candidemia were BDG ≥ 200 pg/mL (*p* < 0.001), CRP ≥ 130 mg/L (*p* < 0.001), and PCT < 3 ng/L (*p* = 0.012); PSEP > 700 pg/mL was borderline discriminative (*p* = 0.091). Sensitivity, specificity, PPV, and NPV values for each of these biomarkers are shown in Table 3. PSEP exhibited greater diagnostic accuracy than positive PCT (<3 ng/L, 76%) and positive CRP (≥5 mg/mL, 100%).

To find markers discriminating between catheter surface *Candida* spp. colonization and sepsis, serum BDG concentrations were analyzed in three patient groups with candidemia confirmed by peripheral blood culture: those without catheter colonization (group 1), those with catheter-related candidemia, i.e., catheter colonization predating IBC (group 2), and those with probable deep-seated candidiasis, i.e., patients in whom catheter colonization was confirmed by culture 1–3 days after IBC (group 3; see Figure 2). *Candida* cultures from catheters were grown within an interval of +/− 3 days relative to IBC when the patient’s CVC or PICC was replaced (Appendix A). Patients with catheter-related candidemia had a significantly lower median BDG concentration (471 pg/mL, *p* < 0.001) than those with probable deep-seated candidiasis (1029 pg/mL) or catheter colonization confirmed after a positive blood *Candida* culture (1203 pg/mL).

### 3.4. Monitoring of Successful Echinocandin Therapy

Appendix A shows that the marker exhibiting the most significant change after initiating echinocandin therapy was PSEP, whose concentration fell significantly (*p* = 0.0012, Wilcoxon signed-rank test) within 14 days (Figure 3). Conversely, serum BDG and CRP concentrations only decreased significantly 28 days after initiating echinocandin therapy (*p* = 0.0038 and *p* = 0.03, respectively). The concentration of PCT did not change over the 28 days in the candidemia group.

### 3.5. Proposal for the Use of Results in Clinical Practice

We propose that serum BDG measurements acquired within 3 days before IBC could be used to distinguish between catheter-related *Candida* sepsis and candidemia from other sources (Figure 2). Moreover, serum BDG concentrations > 200 pg/mL predict invasive candidiasis with high specificity and PPV. For preliminary IC diagnosis, measurement of the inflammatory serum biomarker PSEP using a cut-off of 700 pg/mL seems to be promising (Table 2 and Table 3). In patients with two or more risk factors for IC (Figure 4) together with BDG > 200 pg/mL and PSEP > 700 pg/mL, diagnostic algorithm can predict IC before obtaining a positive blood culture. Possible catheter-related sepsis is predicted by having a CVC together with serum BDG concentrations above 200 pg/mL (median: 471 pg/mL) but below values associated with IC. High BDG concentrations (median 1029 pg/mL) in patients with suspected intraabdominal sepsis (anastomotic leak, postoperative abscess, repeated surgery for recurrent abdominal sepsis or pancreatitis) suggests deep-seated candidiasis.

## 4. Discussion

### 4.1. PCT and PSEP in Sepsis and Sepsis-Related Mortality

Recent publications suggest that PCT and PSEP are the most accurate biomarkers for diagnosing sepsis. A systematic review and meta-analysis of 19 studies on the diagnostic value of PCT and PSEP for sepsis in critically ill adults suggested that these biomarkers have similar diagnostic accuracy, with AUC values of 0.84 and 0.87, respectively [20]. A separate systematic review of 16 studies collectively including 45,079 patients and 785 cases of candidemia indicated that PCT should not be used on its own to discriminate between candidemia and bacteremia due to its low reliability for guiding therapy [21]; it was concluded that PSEP had better prognostic potential. Accordingly, there is a strong positive correlation between PSEP and SOFA scores [11], whereas the corresponding correlations for PCT concentrations are low to negative [7,8]. Our results are consistent with these reports. Furthermore, PSEP exhibited a higher predictive value than PCT concerning IC mortality (AUC = 0.74 vs. AUC = 0.31) and showed better diagnostic performance in cases of IC when using PSEP cut-off concentrations of both 350 pg/mL and 700 pg/mL, giving correct diagnoses in 58/58 patients in both cases.

However, the accuracy of sepsis diagnosis based on PCT was reported to be significantly higher than diagnosis based on PSEP in patients with acute kidney injury (AKI) [19]. Critically ill AKI patients mainly suffer from bacterial sepsis. In IC, however, PCT performs less well; PSEP could fill this diagnostic gap [7,11]. 

### 4.2. PSEP and Renal Function

Presepsin is a relatively small molecule with a molecular weight of 13 kDa that is filtered by the glomeruli and subsequently reabsorbed and proteolyzed in the proximal tubules of the kidney. It would mean that patients with a reduced glomerular filtration rate due to acute renal insufficiency would have higher serum presepsin concentrations. AKI can affect the diagnostic accuracy of PSEP.

Because presepsin is a valuable diagnostic and prognostic biomarker, several studies addressed its connection to sepsis development [19,22,23,24,25]. A range of presepsin levels in non-septic patients was established depending on renal function for individuals with normal renal function and renal failure. If we divided patients due to glomerular filtration rate to five categories: G1 GFR ≥ 90 ml/min/1.73 m^3^, G2 GFR 60–90 ml/min/1.73 m^3^, G3 GFR 30–60 ml/min/1.73 m^3^, G4 GFR 15–30 ml/min/1.73 m^3^, and G5 GFR ≤ 15 ml/min/1.73 m^3^, with reduced GFR, the median of presepsin increases as follows: G1 + G2: 69.8 (60.8–85.9) pg/mL, G3: 107 (68.7–150.0) pg/mL, G4: 171 (117.0–200.0) pg/mL, G5: 251 (213.0–297.5) pg/mL [24]. From this point of view, problems are severe forms of renal failure (grades 4, 5), as reported by Nagata et al., who demonstrated that the presepsin baseline levels were significantly higher in grade 4 and 5 renal failure, achieving 320.2 ± 170 pg/mL and 712.8 ± 336.3 pg/mL, respectively [26]. 

The commonly reported median PSEP concentration is close to 700 pg/mL [19] and the lowest concentrations of PSEP in fungal sepsis are also reportedly close to 700 pg/mL [11]. Data show that 100% of candidemia patients without AKI tested positive for serum PSEP when using a cut-off value of 350 pg/mL, and 100% of patients with stage 2 AKI [according to the Kidney Disease Improving Global Outcomes KDIGO criteria [27] tested positive when using a cut-off value of 700 pg/mL; stage 2 AKI predominated among our AKI patients (Table 1 and Table 2). Thus, it appears that PSEP levels can be a reliable indicator of sepsis in patients with non-injured or moderately injured kidneys (Stages 1–2, Table 1). A PSEP cut-off of 700 pg/mL may be optimal in AKI stages 1 and 2. These conclusions are consistent with previous reports [19,22,24]. They are supported by a study on the diagnostic accuracy of PCT and PSEP for infectious diseases in AKI patients, suggesting that both biomarkers are helpful, albeit with different thresholds [23,25,28].

Renal dysfunction is a common complication of sepsis; this is often the first manifestation of evolving multiorgan dysfunction syndrome (MODS). Renal replacement therapy (RRT) also has its place in non-renal indication in treating patients with septic shock. The PSEP molecular weight of 13 kDa is slightly higher than that of β2-microglobulin (β2–MG, 11.8 kDa). The elimination of presepsin was confirmed using a dialysis membrane with clearance β2-MG ≥ 50 ml/min (for a 2nd-hour removal ratio of 42.8 ± 7.9%, and a 4th-hour removal ratio of 58.8 ± 18.4%) [29]. These methods of RRT improve presepsin filtration. The onset of sepsis even after RRT initiation remains a clinical challenge. The cohort of patients in this study (58 subjects) was monitored for renal functions and RRT needs. Therefore, it was possible to exclude patients with severe AKI from calculating the cut-offs presented in the diagnostic algorithm (Figure 4). 

### 4.3. Predictive Value of Biomarker Combinations in Invasive Candidiasis

It is broadly accepted that the diagnosis of invasive *Candida* infections, which are a major cause of sepsis, should be based on simultaneous determination of multiple biomarkers including early inflammatory markers (CRP, PCT, and PSEP) and at least one highly specific biomarker, such as the panfungal antigen BDG [4,30,31]. The presented study confirmed that pre-existing bacterial colonization in candidemia patients had little effect on false BDG positivity (0–224 pg/mL). The ICU patients are very heterogeneous. They are recovering from major abdominal surgery, having BDG concentrations > 100 pg/mL associated with significantly increased SOFA scores and mortality ranging from 13.7% (BDG ≤ 100 pg/mL) to 39.0% (>100 pg/mL) [32]. Our results confirm the diagnostic value of BDG at serum concentrations > 250 pg/mL and particularly at borderline concentrations > 200 pg/mL in IC patients [31].

BDG plays an essential role in predicting IC because early inflammatory biomarkers are insufficiently specific to distinguish *Candida* from bacterial sepsis [20,21]. Although our data suggest that CRP has great potential for distinguishing between candidemia and bacteremia at a cut-off of 130 mg/L (*p* = 0.005), its sensitivity and specificity for *Candida* sepsis are poor (67.2% and 66.9%, respectively; AUC = 0.67). A recent analysis showed that combining CRP with other biomarkers can increase sensitivity or specificity for *Candida* sepsis (AUC = 0.912) [33]. However, CRP is not an appropriate biomarker for guiding antifungal therapy, unlike PCT for antibacterial treatment [34,35,36] or BDG for antifungal therapy [37].

In the presence of risk factors for IC, diagnosing both bloodstream and deep-seated candidiasis requires combined tests [38]. The negative predictive value of BDG for IC is well evidenced [39,40], and BDG has been shown to influence treatment decisions [41]. However, the clinical significance of elevated BDG concentrations in critically ill patients remains unclear [6]. The BDG is a panfungal antigen with diverse concentrations in heterogeneous groups of IC patients, including those with bloodstream candidiasis, *Candida* sepsis, and deep-seated candidiasis. This work analyzed samples from 58 patients with IC, finding statistically significant differences among these IC forms.

### 4.4. PSEP, Monitoring, and Prediction of Successful Treatment

In contrast to the limited evidence concerning the predictive value of PCT for monitoring the effect of antifungal or antibiotic therapy [21,34,36], the high predictive value of serum PSEP for evaluating the success of echinocandin therapy in ICU patients with candidemia was demonstrated. A statistically significant connection between decreased serum PSEP concentrations and successful treatment became apparent within 14 days of initiating therapy in contrast to BDG or CRP, which can indicate the therapeutic response 28 days after antifungal therapy initiation (Figure 3). Our findings in 17 patients expand on an earlier study of 7 patients reported by Bamba et al., who found that PSEP levels fell rapidly in successfully treated patients but rose continuously in poorly responding ones [11]. Together, these results identify PSEP as an important predictor of the success of echinocandin therapy, which is recommended as first-line treatment for IC and candidemia, with de-escalation to fluconazole when clinical stability is achieved [31,38].

### 4.5. Proposal for the Use of Results in Clinical Practice

Figure 4 outlines a diagnostic algorithm for non-neutropenic and non-transplanted ICU patients at risk of IC and/or candidemia. This algorithm extends one previously developed for ICU patients with suspected candidemia in sepsis of abdominal [31] and non-abdominal origin [4]. Our extended algorithm can accommodate both forms of sepsis, and the combination of PSEP and BDG could greatly increase the specificity of non-culture based methods for early diagnosis of IC in ICUs (Table 3; SN = 94.8%, SP = 100%, PPV = 100%, NPV = 97.3% with AUC = 0.97). Moreover, it may be helpful in cases of probable IC [12]. Using this study, it was possible to obtain threshold concentrations of BDG (>200 pg/mL) and PSEP (>700 pg/mL), which can serve as a guide in distinguishing the probable origin of IC, such as deep-seated candidiasis from abdominal one (BDG > 1029 pg/mL), or catheter candida sepsis (BDG > 471 pg/mL). Subsequently, PSEP can be used as a tool to assess the effectiveness of selected IC therapy.

## 5. Limitations

The main limitation of our study is its retrospective nature. In patients, only creatinine levels were evaluated; diuresis was not taken into account due to the study’s retrospective nature as diuresis over 6–12 hours is tough to determine retrospectively. Consequently, some of the measured values may deviate from those at the time of the index blood culture. However, this effect should be minimized because we mainly included patients whose BDG, PSEP, PCT, and CRP levels had been determined shortly before or after the positive index blood culture. It should also be noted that PSEP levels are sensitive to the stage of AKI and dialysis after AKI. However, this effect may be attenuated using a PSEP cut-off concentration of 700 pg/mL. 

The small number of consecutive patients limited the analysis of responses to echinocandin therapy included in the study. Due to the high incidence of candidiasis in the population, sensitivity and specificity could be biased. The final limitation includes excluding neutropenic and transplanted patients because their immunodeficiency could decrease the PSEP cut-off detected in our patient’s cohort. Therefore, the applicability of PSEP detection in severely immunocompromised patients remains to be tested in a future study.

## 6. Conclusions

Combined analysis of PSEP and BDG is a potent tool for differential diagnosis and determining the onset of invasive candidiasis in critically ill patients.

The use of PSEP and BDG could be helpful in the diagnostic workflow for critically ill patients with probable invasive candidiasis and for guiding antifungal therapy, including highlighting cases where early initiation is warranted and choosing a duration and de-escalation. The advantage of PSEP as a biomarker is that its concentration can be determined within 15 minutes using the Pathfast method. However, further studies are needed to determine whether this could significantly impact the use of PSEP and BDG in the diagnostic algorithm for non-neutropenic and non-transplanted ICU adults at risk for invasive candidiasis and/or candidemia.

## Figures and Tables

**Figure 1 jof-08-00308-f001:**
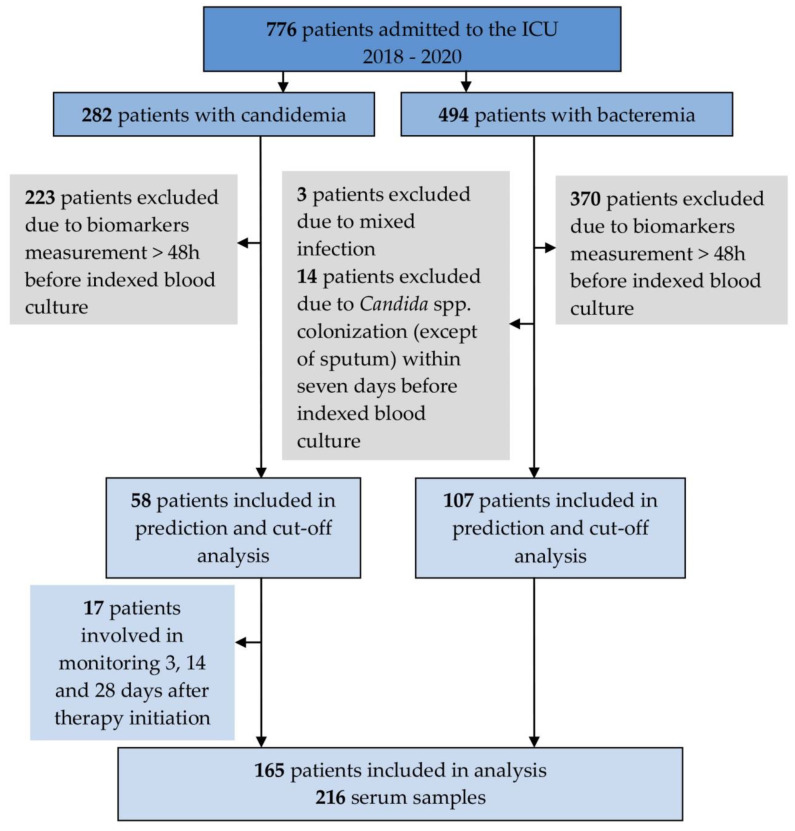
Flowchart showing the process used to select patients for inclusion in the analysis.

**Figure 2 jof-08-00308-f002:**
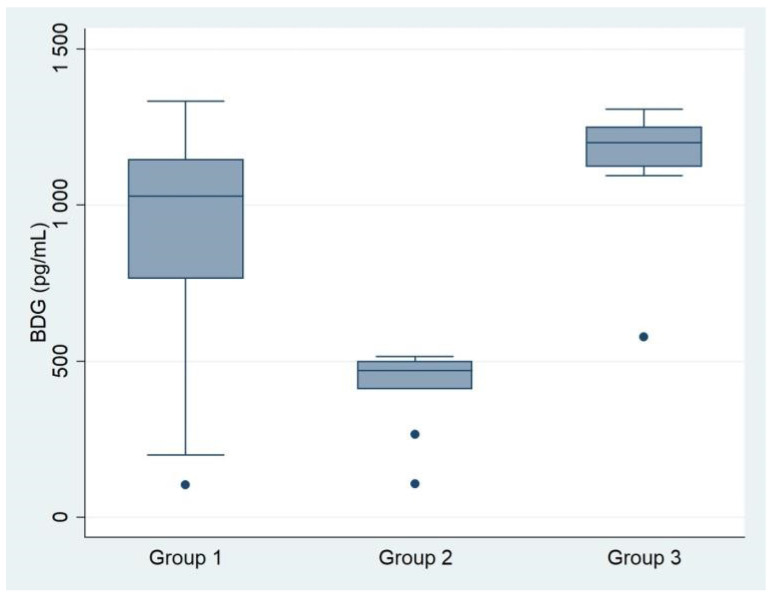
Serum BDG concentrations in three patient groups with candidemia confirmed by blood cultures, differing in the timing of culture-confirmed catheter surface candidiasis. Group 1: Patients without catheter colonization (i.e., negative catheter culture on the day of IBC), indicating that invasion preceded catheterization. Median BDG concentration: 1029 pg/mL. Group 2: Candidemia detected after culture-confirmed catheter colonization (catheter-related candidiasis; catheter culture was established 1–3 days before confirmation of candidemia). Median BDG concentration: 471 pg/mL. Group 3: Catheter colonization confirmed 1–3 days after proof of candidemia by culture. Median BDG concentration: 1203 pg/mL.

**Figure 3 jof-08-00308-f003:**
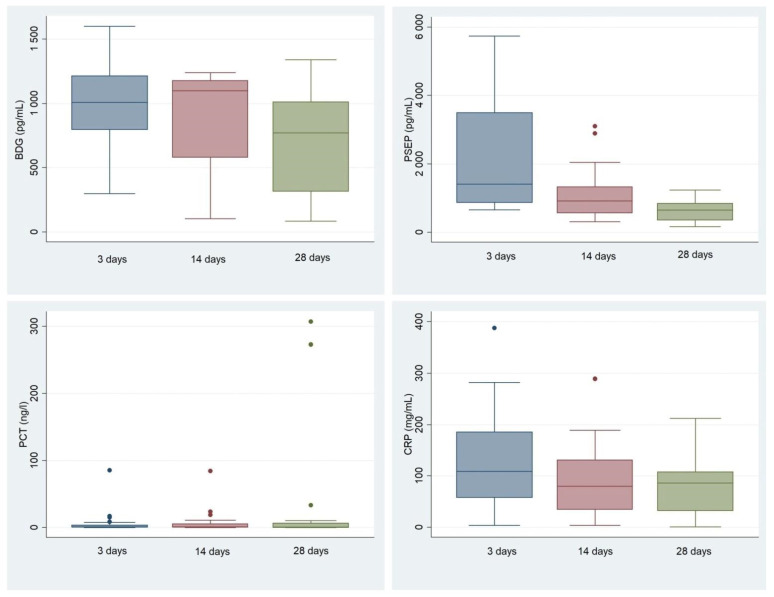
Changes in the *Candida* sepsis biomarkers PCT, PSEP, CRP, and 1,3-β-D-glucan within 3, 14, and 28 days after echinocandin (ATM) therapy.

**Figure 4 jof-08-00308-f004:**
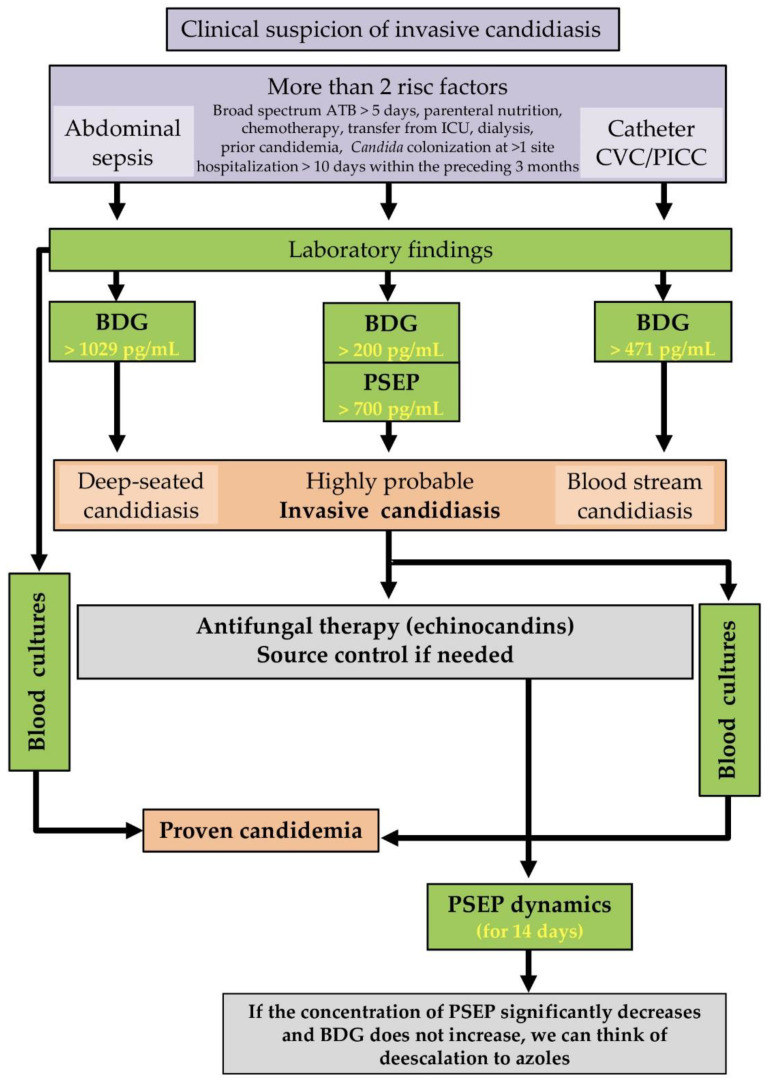
Use of PSEP and BDG serum concentration data in a diagnostic algorithm for non-neutropenic and non-transplanted ICU patients at risk of IC and/or candidemia. BDG, 1,3-β-D-glucan; PSEP, presepsin; BSC, bloodstream candidiasis (candidemia); DSC, deep-seated candidiasis; CS, catheter sepsis; Abdominal sepsis refers to an anastomosis leak, postoperative abscess, repeated surgery for recurrent abdominal sepsis, or infected pancreatitis.

**Table 1 jof-08-00308-t001:** Patients’ characteristics (n = 165). *p*-values indicate the significance of the difference between the candidemia and bacteremia groups for each variable at the time of infection.

Patients’ Characteristics	Candidemia * (n = 58; 35%)	Bacteremia ** (n = 107; 65%)	*p*-Value
**Demographic**			
Median age (IQR)	54 (46–72)	60 (46–72)	0.668 ^†^
Male gender	34 (58%)	76 (71%)	0.116 ^✕^
Mortality rate	19 (33%)	24 (22%)	0.141 ^♢^
**Clinical type of patients**	n (%)	n (%)	
Surgical	19 (33%)	25 (23%)	0.193 ^†^
Traumatic	2 (3%)	6 (6%)	0.714 ^♢^
Medical	37 (64%)	76 (71%)	0.340 ^†^
Previous abdominal surgery	14 (24%)	13 (12%)	0.047 ^†^
Diabetes mellitus	1 (2%)	0 (0%)	0.352 ^♢^
Inserted CVC or PICC	58 (100%)	74 (69%)	<0.001 ^†^
Chronic renal disease	12 (21%)	12 (11%)	0.099 ^†^
Corticosteroid therapy	10 (17%)	14 (13%)	0.470 ^†^
Preceding antibiotic therapy	48 (83%)	45 (42%)	<0.001 ^†^
Preceding antifungal therapy	6 (10%)	7 (6.5%)	0.387 ^†^
**Prescoring**	median (IQR)	median (IQR)	
SOFA	3.5 (2–9)	6.0 (2–8)	0.191 ^✕^
APACHE II	12.5 (8–18)	15.0 (10–21)	0.200 ^✕^
Time from admission to candidemia or bacteremia, days	23.0 (18–30)	13 (10–18)	<0.001 ^✕^
**Acute kidney injury stage (serum creatinine range)**	n = 18 (31%)	n = 40 (33%)	
Stage 1 (110–170 µmol/L)	4 (22%)	16 (40%)	
Stage 2 (171–299 µmol/L)	11 (61%)	17 (43%)	0.362 ^†^
Stage 3 (300 ≥ 440 µmol/L)	3 (17%)	7 (17%)	
Median serum creatinine, µmol/L (IQR)	274 (178–348)	180 (141–321)	0.217 ^✕^
CRRT	3 (5%)	3 (3%)	
RRT	10 (17%)	12 (11%)	

Values are shown as n (%), ^†^ Pearson’s chi-squared, ^✕^ Mann–Whitney two-sample test, ^♢^ Fisher’s exact test. * *Candida albicans* (n = 24), *C. tropicalis* (n = 11), *C. krusei* (n = 7), *C. glabrata* (n = 6), *C. parapsilosis* (n = 4), *C. dubliniensis* (n = 1), *C. guilliermondii* (n = 1), *C. lusitaniae* (n = 1), *C. metapsilosis* (n = 1), *Saccharomyces cerevisiae* (n = 1), *Geotrichum clavatum* (n = 1). ** coagulase-negative staphylococci (n = 39), *Enterococcus* spp. (n = 13), *Pseudomonas aeruginosa* (n = 10), *Klebsiella pneumoniae* (n = 9), *Escherichia* spp. (n = 6), *Acinetobacter* spp. (n = 5), *Propionibacterium acnes* (n = 4), *Enterobacter* spp. (n = 3), *Micrococcus luteus* (n = 3), *Staphylococcus aureus* (n = 3), *Stenotrophomonas maltophilia* (n = 3), *Streptococcus* spp. (n = 2), *Bacillus cereus* (n = 2), *Proteus penneri* (n = 1), *Serratia marcescens* (n = 1), *Actinomyces odontolyticus* (n = 1), *Burkholderia multivorans* (n = 1), *Corynebacterium* spp. (n = 1). CVC, central venous catheter; PICC, peripherally inserted central catheter; IQR, interquartile range; CRRT, Continuous Renal Replacement Therapy; RRT, Renal Replacement Therapy.

**Table 2 jof-08-00308-t002:** Prediction and differential diagnosis of *Candida* and bacteria sepsis using biomarkers at different cut-off values.

Assessed Biomarkers (Recent Cut-Off)	Candidemia n (%)	Bacteremia n (%)	*p*-Value
CRP (≥5 mg/mL)	58 (100%)	107 (100%)	0.999 ^†^
CRP (≥130 mg/mL)	22 (38%)	73 (68%)	<0.001 ^†^
PCT (>0.5 ng/l)	52 (90%)	91 (85%)	0.406 ^†^
PCT (<3 ng/l) *	44 (76%)	60 (56%)	0.012 ^†^
PSEP (>350 pg/mL)	58 (100%)	107 (100%)	0.999 ^✕^
PSEP (>700 pg/mL) **	58 (100%)	101 (94%)	0.091 ^✕^
BDG (≥80 pg/mL)	58 (100%)	19 (18%)	<0.001 ^✕^
BDG (≥200 pg/mL)	56 (97%)	3 (3%)	<0.001 ^†^
**Median Values (Conc.)**	**Candidemia (IQR)**	**Bacteremia (IQR)**	***p*-Value**
CRP (mg/L)	104 (78–150)	164 (101–234)	<0.001 ^♢^
PCT (ng/L)	1.6 (0.90–2.80)	2.4 (1.03–8.54)	0.105 ^♢^
PSEP (pg/mL)	1784 (1203–3259)	1963 (1313–3524)	0.777 ^♢^
BDG (pg/mL)	1029 (500–1176)	35 (0–73)	<0.001 ^♢^

Some values are shown as n (%), ^†^ Pearson’s chi-squared, ^✕^ Fisher’s exact test, ^♢^ Mann–Whitney two-sample test. The CRP, PCT, PSEP, and BDG biomarkers’ positivity was expressed relative to preselected cut-offs (in accordance with * and ** below). CRP, C-reactive protein; PCT, procalcitonin; PSEP, presepsin; BDG, 1,3-β-D-glucan; conc., concentration. * The concentration of PCT in IC was in the 2–3 ng/L in the range [7,8]. Hence, concentrations < 3 ng/mL indicate *Candida* sepsis with high probability. ** Acute stage 2 kidney injury can affect the diagnostic accuracy of PSEP; its median serum concentration is usually close to 700 pg/mL [19] and the lowest concentrations of PSEP in fungal sepsis are also generally close to 700 pg/mL [11].

**Table 3 jof-08-00308-t003:** Biomarker cut-offs with the highest diagnostic significance for invasive candidiasis.

	CRP 130 mg/mL	PCT 0.5 ng/L	PCT 0–3 ng/L	PSEP >700 pg/mL	BDG ≥200 pg/mL	BDG/PSE *
Sensitivity (%)	37.9	89.7	75.9	100	96.6	94.8
95% CI	25.5–51.6	78.8–96.1	62.8–86.1	93.8–100	88.1–99.6	85.6–98.9
Specificity (%)	31.8	15.0	43.9	5.6	97.2	100
95% CI	23.1–41.5	8.8–23.1	34.3–53.9	2.1–11.8	92.0–99.4	96.6–100
PPV (%)	23.2	36.4	42.3	36.5	94.9	100.0
95% CI	15.1–32.9	28.5–44.8	32.7–52.4	29.0–44.5	85.9–98.9	93.5–100.0
NPV (%)	48.6	72.7	77.0	100	98.1	97.3
95% CI	36.4–60.8	49.8–89.3	64.5–86.8	54.1–100	93.4–99.8	92.2–99.4
AUC	0.35	0.52	0.60	0.53	0.97	0.97
95% CI	0.27–0.43	0.47–0.58	0.53–0.67	0.51–0.55	0.94–1.00	0.95–1.00

* BDG/PSEP—if both tests are positive, i.e., BDG ≥ 200 pg/mL and PSEP > 700 pg/mL.

## Data Availability

All data analyzed during the current study are included in this article and its Appendix A.

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
