# Peer review of "Combined Use of Presepsin and (1,3)-β-D-glucan as Biomarkers for Diagnosing Candida Sepsis and Monitoring the Effectiveness of Treatment in Critically Ill Patients"

_jof, 2022, doi:10.3390/jof8030308_

Round 1

Reviewer 1 Report

This is a well structured retrospective and multicentric study with valuable results. However, Revision of the writing style, methods and dicussion is needed.

- The writing style in general needs improvement (for example repetitive use of words, excessive use of "we"),proper language check is recommended.

- The choice of those cutof for each marker should be highlited in methods.

- The number of patients on dialysis and frequency of dialysis was not clearly taken into concideration, although the authors stated that PSEP can be reduced by dialysis. Otherwise, ruling out patients with AKI or at least on dialysis should be concidered in order to improve the reliability of the results.

- In table 3 the coloring is confusing and if it is vital then it should be explaned in legend.

- Are there other tables that are not provided? where ist Table ST1? if it refers to a part of table 1 then it should be explaned better, ist not comprehensive in the form.

- The discussion must be improved, basically the results are pointed out here more that being discussed.

Author Response

Dear reviewers:                                                                                                Ostrava, Feb 23, 2022

We are grateful for the overall positive tone of the three review reports. Below, please find our point-by-point responses to the reviewers' comments. The manuscript modifications have been performed in MS Word track changes mode. Thank you and all reviewers for critical reading, comments, and recommendations. Thanks to the criticisms, we could improve our manuscript both in clarity and fluency. Please, express our gratitude to the referees.

With kind regards,

Marcela Káňová, on behalf of all contributors

Reviewer comments:

Reviewer #1:

- The writing style in general needs improvement (for example repetitive use of words, excessive use of "we"), proper language check is recommended.

R: The final version of the manuscript was edited by a qualified English-speaking expert.

- The choice of those cut-off for each marker should be highlighted in methods.

R: Done. 

- The number of patients on dialysis and frequency of dialysis was not clearly taken into consideration, although the authors stated that PSEP can be reduced by dialysis. Otherwise, ruling out patients with AKI or at least on dialysis should be considered in order to improve the reliability of the results.

R:  We partly agree with this reviewer. AKI is a considerable concern in patients with sepsis or septic shock; kidneys are among the first failing organs during the development of multiorgan dysfunction. In addition, there is a non-renal indication for RRT in septic shock therapy. So, it is not possible to rule out all these patients. On top of that, the relationship between presepsin and renal failure has been extensively studied. We followed the results of recent meta-analyses indicating that presepsin could be used as a valuable early biomarker of sepsis, even with its inherent limitations. On the other hand, patients with severe renal failure were ruled out. Table 1 was supplemented with dialysis types with the appropriate patient representation, and the discussion is enhanced with the relationship of PSEP to all AKI types.

- In table 3 the coloring is confusing and if it is vital then it should be explained in legend.

R: We removed the colors.

- Are there other tables that are not provided? Where is Table ST1? If it refers to a part of table 1 then it should be explained better, is not comprehensive in the form.

R: The supplementary material is now well organized and referenced in the main text.

- The discussion must be improved, basically the results are pointed out here more that being discussed.

R: The discussion was improved – please, see the changes in colors.

Reviewer 2 Report

This is a multicenter retrospective observational cohort study in adult ICU patients.

The study tried to assess the clinical significance of the use of biomarkers for the early diagnosis of fungal or bacterial sepsis and for monitoring the therapeutic outcome. The authors did so by identifying suitable cut-off levels at certain time-points based on biomarkers’ kinetics.

The prognostic value of repeated measurements of serum biomarkers, such as  (1,3)-β-D-glucan alone or combined with other inflammatory biomarkers has been extensively studied to date in cases of candidemia and/ or candidiasis in critically ill patients in the adult or neonatal population.

Presepsin is a relatively novel biomarker for sepsis; little is known about its usefulness in invasive fungal infection and associated-sepsis.

Given that candidemia is a major cause of bloodstream infection in tertiary hospitals worldwide, fungal biomarkers might provide a prognostic model of early diagnosis of fungal infections and monitoring of the therapeutic efficacy.

Comments for consideration:

-Line 108: was part of a registered..

-Line 118: the authors say “Patients with cultures positive for Candida spp. and bacteria more than 48 h apart were excluded”; why did you exclude those episodes of candidemia or bacteremia? How were you sure it was not a subsequent episode of nosocomial infection in the same patient? Please comment.

-In line 112-113 you say “(a) BDG within 48 h before the time of the indexed blood culture (IBC) result, and (b) PCT, CRP and PSEP within 24 h before IBC”, whereas, in lines 148-49 you say “consecutive measurements of biomarkers were performed on days 3, 14, and 28 after initiating echinocandin therapy”; please amend the above lines (112-3).

-Study period is January 2018 and December 2020. In figure 1 first box, you say 2018-9; please correct.

-In the variables in table 1, the authors should add, depth/duration of neutropenia, TPN, subgroup of medical patients; immunocompromised or not, i.e., hematological malignancy, chemotherapy etc

-In the limitation paragraph the authors should add the absence of neutropenic or bone/ solid-organ transplant patients, and should comment that a clinical study in neutropenic patients is warranted to prove if such biomarkers still can be used in severely immunocompromised patient population.

-In the conclusive paragraph of discussion (4.4) the authors should briefly summarize further the key-points from paragraph 3.5. (“Proposal for the use of results in clinical practice”).

-Lines 240-2 require re-writing; it is a misinterpretation when they say that CRP is ‘best biomarker of candidemia’. Quite the opposite; a high CRP more often is linked to bacterial sepsis.

-Figure 4 is an interesting diagnostic algorithm but not easy to read! Try to simplify it for the average reader. Thank you.

Author Response

Dear reviewers:

We are grateful for the overall positive tone of the review reports. Below, please find our point-by-point responses to the reviewers' comments. The manuscript modifications have been performed in MS Word track changes mode. Thank all reviewers for critical reading, comments, and recommendations. Thanks to the criticisms, we could improve our manuscript both in clarity and fluency. Please, express our gratitude to the referees.

With kind regards,

Marcela Káňová, on behalf of all contributors

Reviewer comments:

Reviewer #2:

-Line 108: was part of a registered...

R: The statement was corrected.

-Line 118: the authors say "Patients with cultures positive for Candida spp. and bacteria more than 48 h apart were excluded"; why did you exclude those episodes of candidemia or bacteremia? How were you sure it was not a subsequent episode of nosocomial infection in the same patient? Please comment.

R: Corrected. We excluded only those episodes of candidemia or bacteremia, where the data (biomarker versus blood cultures) did not overlap according to the biomarkers' kinetics to improve the reliability of results.

-In line 112-113 you say "(a) BDG within 48 h before the time of the indexed blood culture (IBC) result, and (b) PCT, CRP and PSEP within 24 h before IBC", whereas, in lines 148-49 you say "consecutive measurements of biomarkers were performed on days 3, 14, and 28 after initiating echinocandin therapy"; please amend the above lines (112-3).

-Study period is January 2018 and December 2020. In figure 1 first box, you say 2018-9; please correct.

R: The flow chart in Figure 1 was corrected.

-In the variables in table 1, the authors should add, depth/duration of neutropenia, TPN, subgroup of medical patients; immunocompromised or not, i.e., hematological malignancy, chemotherapy etc

R: Neutropenic and transplanted patients were excluded due to the small subcohort size. The chapter Material and Methods was modified accordingly.

-In the limitation paragraph the authors should add the absence of neutropenic or bone/ solid-organ transplant patients and should comment that a clinical study in neutropenic patients is warranted to prove if such biomarkers still can be used in severely immunocompromised patient population.

R: The Limitations chapter was added.

-In the conclusive paragraph of discussion (4.4) the authors should briefly summarize further the key-points from paragraph 3.5. ("Proposal for the use of results in clinical practice").

R: We amended Chapter 4.5 (after renumbering) with a conclusive statement.

-Lines 240-2 require re-writing; it is a misinterpretation when they say that CRP is 'best biomarker of candidemia'. Quite the opposite; a high CRP more often is linked to bacterial sepsis.

R: Corrected: the sentence in question was modified.

-Figure 4 is an interesting diagnostic algorithm but not easy to read! Try to simplify it for the average reader. Thank you.

R: Figure 4 was modified.

Reviewer 3 Report

This multicenter retrospective study aims to develop a new biomarker panel, based on serum levels of the panfungal biomarker (1,3) -β-D-glucan (BDG) and the inflammatory biomarkers C-reactive protein, presepsin (PSEP), and procalcitonin, to enable more accurate invasive candidiasis diagnosis. The study enrolled 58 patients with candidemia and 107 patients with bacteremia. The process used to select patients for inclusion in the analysis was accurate and appropriate. Although some data were expected and confirmed what is already known in the literature, there are some ideas for novelty. The authors suggest that combined analysis of PSEP and BDG is a  powerful tool for differential diagnosis and determining the onset of invasive candidiasis in critically ill patients and indicate the high predictive value of serum PSEP for evaluating the success of echinocandin therapy in ICU patients with candidemia.

Author Response

Dear reviewer:                                                                                                Ostrava, Feb 23, 2022

We are grateful for the overall positive tone of the three review reports. Below, please find our point-by-point responses to the reviewers' comments. The manuscript modifications have been performed in MS Word track changes mode. Thank you and all reviewers for critical reading, comments, and recommendations. Thanks to the criticisms, we could improve our manuscript both in clarity and fluency. Please, express our gratitude to the referees.

With kind regards,

Marcela Káňová, on behalf of all contributors